# Change in Values of Illegal Miners and Inhabitants and Reduction in Environmental Pollution after the Cessation of Artisanal and Small-Scale Gold Mining: A Case of Bunikasih, Indonesia

**DOI:** 10.3390/ijerph20176663

**Published:** 2023-08-28

**Authors:** Idham Andri Kurniawan, Win Thiri Kyaw, Mirzam Abdurrachman, Xiaoxu Kuang, Masayuki Sakakibara

**Affiliations:** 1Geological Engineering Department, Faculty of Earth Sciences and Technology, Bandung Institute of Technology, Bandung 40132, Indonesia; mirzam@gl.itb.ac.id; 2Research Institute for Humanity and Nature, Kyoto 603-8047, Japan; thiri@chikyu.ac.jp (W.T.K.); sakakibara.masayuki.mb@ehime-u.ac.jp (M.S.); 3Chaozhou Branch of Chemistry and Chemical Engineering Guangdong Laboratory, Chaozhou 521000, China; xxkuang@chikyu.ac.jp; 4Faculty of Collaborative Regional Innovation, Ehime University, Matsuyama 790-8577, Japan

**Keywords:** ASGM, transdisciplinary community of practice, tea leaves, mercury, lead, arsenic, Bunikasih, Indonesia

## Abstract

Artisanal and small-scale gold mining (ASGM) harms the environment and human health, because it requires mercury (Hg). Therefore, this study uses a questionnaire survey to identify the effects of the medical checkup findings, group discussion, and educational seminar on the values of illegal miners and residents in Bunikasih, Indonesia, regarding the environmental and health effects of Hg. Transdisciplinary communities of practice (TDCoP) were formed to pursue alternative livelihoods than illegal ASGM. Environmental pollution after ASGM closure was assessed. The questionnaire showed that respondents changed their views on Hg’s effects and were willing to stop ASGM practices. In an environmental impact assessment study, tea leaf samples were obtained between the ASGM activity location and village housing in two periods during ASGM closure to identify the concentrations of Hg, lead, and arsenic. Their mean concentration values in Period 1 (2) were 0.54 ± 0.14, 0.66 ± 0.09, and 0.34 ± 0.12 mg kg^−1^ (0.08 ± 0.04, 0.34 ± 0.14, and 0.07 ± 0.06 mg kg^−1^), respectively, indicating a decrease in environmental pollution. In conclusion, the government-driven cessation of ASGM in the study area shows a reduction in environmental pollution, and the change in the perception of the participants regarding the ASGM and TDCoP approaches help to make these changes permanent.

## 1. Introduction

### 1.1. Artisanal and Small-Scale Gold Mining (ASGM) in Indonesia

Artisanal and small-scale gold mining (ASGM) is a major source of mercury (Hg) pollution worldwide [1]. The ASGM process is the fastest and most economical way of refining gold (Au) [2]. However, elemental Hg is used in this process, making it also the most dangerous Au-refining process, because Hg contamination can quickly reach surrounding living organisms or spread widely through air, water, and soil [3]; eventually, these contaminants settle in the sediment of lakes, rivers, or estuaries, where it is converted into methylmercury (MeHg), which further enters into the food chain. Elemental Hg and MeHg are neurotoxic; the Hg vapor can damage the nervous system, digestive system, immune system, lungs, and kidneys, which can be lethal [1]. ASGM activities, specifically their mineral deposits and anthropogenic factors, produce heavy metal pollutants apart from Hg, such as arsenic (As), lead (Pb), and cadmium (Cd) [4,5,6], which also cause detrimental health effects.

ASGM activities are estimated to contribute to the release of 57.5% of the total national Hg emission in Indonesia [7]. The high level of Hg contamination in the country is the result of excessive thermal decomposition of amalgams in open pans and, more importantly, the amalgamation of the whole ore in small ball mills, known locally as “tromols”. To produce 1 g of gold, 10–25 g of Hg is needed for whole ore amalgamation, whereas the concentration methods need only 1–3 g of Hg [8]. The ratio of Hg lost to Au produced (i.e., the Hg lost per unit of gold) in Asian and Latin American countries using the whole ore amalgamation practice is between 4 and 5, whereas that in Africa ranges between 1 and 2 [9]. As in many other countries in Latin America and Africa, processing facilities proliferated in Indonesia, offering miners amalgamations of ores for free or a nominal fee on the condition that they leave their tailings as payment for the services. Small ball mill amalgamation of gold ores recovers less than 30% of the gold in the ores [10,11]. The Hg emission is almost evenly distributed across the main islands of Indonesia, with over 300,000 miners in more than 1000 ASGM locations in the country [12]. These miners do not have work options other than farming or fishing, from which they do not earn enough.

Various samples, such as air, water, sediment, and soil, have been used to measure the heavy metal pollution caused by ASGM activities [3,5,13,14]. Many factors influence the distribution of pollution, including its natural origin (e.g., rainfall and wind) and human activities (e.g., duration of refining and proper/improper disposal of Hg waste) [3]. Hence, pollution spread, level, and behavior in areas of ASGM activity should be determined.

Government regulations in Indonesia designate Hg as a hazardous and toxic substance (B3). Its use has been prohibited, and various measures have been implemented to prevent its illegal use in mining. The Indonesian government has banned the use of Hg in Au extraction, especially in ASGM activities, ratifying the Minamata Convention on Mercury into domestic law by Law No. 11 of 2017 in September 2017 [12,15,16]. However, these efforts have not been effective, as seen from the many ASGM and other Au extraction activities that still use Hg [17]. Several studies have reported that ASGM contributes to land degradation, river water pollution, and soil/sediment pollution, all of which do not meet the established quality standards [15]. The Indonesian Government Regulation 82/2001 specifies 0.002 mg/L for Hg in river water and 0.001 mg/L for potable water and water supplies. ASGM also causes numerous health defects in miners and people living in mining communities in Indonesia, such as acute and chronic Hg intoxication, lung problems including tuberculosis, and the delayed growth and development of children living near the mines. There is also the possibility of permanent disability and cancer for children in the affected communities [15]. Therefore, addressing the Hg pollution caused by ASGM activities is a major objective that must be fulfilled. Organizations have exerted considerable efforts globally, especially in Indonesia, to stop Hg pollution [12,16,18].

### 1.2. Interventions for the Elimination of Hg from ASGM and Perception of Miners

In response to the need for Hg reduction in ASGM, attempts have been made to solve the problem through environmental and health, technological, and formalization approaches [19]. However, Hg elimination in real ASGM could not be achieved due to the lack of sustainability in having only a limited duration of intervention, inability to deliver the environmental and health impact findings to miners, ineffectiveness in changing the behavior of miners, high cost of techniques, and insufficient resources of authorities [19]. Furthermore, several interventions have been employed in global ASGM sectors, including in Indonesia, by targeting the educational aspects, such as creating awareness about ASGM and the associated risks of Hg exposure and highlighting the effectiveness of cleaner technologies [10,11,20] and alternative techniques, such as using magnets [21], borax [22,23], direct smelting of gold concentrates, and cyanidation. Although some of these attempts could successfully reduce Hg emissions to an extent, their high cost or complexity have limited their practice. In addition, miners believe that gold recovery from alternative methods is poor compared to the amalgamation method. Social factors have also hindered the acceptance of alternative techniques [24].

Educating ASGM miners and the host communities on ASGM and the environmental and health impacts of Hg may influence their perception, which would, in turn, affect their decision making, thereby promoting the implementation of numerous perception-focused educational studies in ASGM [25,26,27,28,29,30]. A survey on Congolese residing near gold mines revealed that the community exhibited a greater level of concern regarding the impacts of deforestation, erosion, river siltation, and acid pollution, but they were relatively less concerned about the effects of Hg and cyanide contamination. This is because these pollutants are not observable; thus, it is challenging to identify them [26]. Those who rely heavily on mining activities are least inclined to sacrifice the immediate economic advantages for long-term environmental and health benefits. Similarly, Charles et al. reported the limited awareness of Hg and As risks [27].

### 1.3. Transdisciplinary Research Practice to Tackle ASGM Issues

Transdisciplinary studies have been conducted to solve the ASGM issues based on the complex nature of poverty and environmental degradation [31]. To facilitate practical transdisciplinary research that involves academic researchers and nonacademic participants, it is vital to consider methodologies that link transdisciplinary research and the uninterrupted participation of nonacademic participants. Additionally, organizational structures are vital to implement the knowledge obtained from such activities [32]. Several communities of practice, as a transdisciplinary community of practice (TDCoP), have been considered for transdisciplinary research to solve complex societal problems. TDCoP was formed to nurture independent communities that can decide, plan, and manage their problems without relying on researchers. The community transforms from participation to collaboration and, finally, to taking autonomous actions.

In the beginning of the TDCoP process, researchers and local communicators lead the process through dialogs with the community to collect basic information and extract problems, along with building mutual trust among members through dialogs [33]. Then, the topic or issue of interest or concern, known as the transformative boundary object (TBO), is identified, which plays a vital role in transformative learning and involves uninterested individuals. Next, the members establish TDCoP. Further, TDCoP members perform basic research, during which the researchers develop the employed technique; the local communicators take on an administrative role and provide connection within the community; and the key stakeholders support the connection with the community, share information, and coordinate the surveys [33]. When the basic research results are obtained, the community and key stakeholders develop transformative learning, where the researchers share scientific knowledge until they perceive a change in values and can narrate the problems with their own words, which, finally, transform them from passive recipients of information to autonomous action-takers. Then, they begin to seek solutions to their problems.

In this study, we combine educational social studies regarding the perception of ASGM miners and communities on ASGM and Hg and sustainable solutions to the implementation of alternative sources of livelihood by cocreating with researchers, communities, and stakeholders through TDCoP. An environmental assessment on the impact of ASGM closure by governments was conducted, which differed from the abovementioned educational studies on perception. This study provides insights into the theory of TDCoPs in solving ASGM problems in Bunikasih, Indonesia.

### 1.4. Preliminary Transdisciplinary Study at the Study Site

Bunikasih is a village in the south of Bandung City, Indonesia, which is located on a tea plantation established during the Dutch Colonial Era. Aside from tea and vegetable farming, the livelihood of Bunikasih Village residents involves illegal ASGM activities.

In March 2019, we conducted a preliminary transdisciplinary study. We evaluated the history and socioeconomic conditions of the area by interviewing stakeholders, such as the village head and inhabitants (including a group of miners). According to the interview results, the illegal ASGM activities in this region started in 1993, after PT Antam’s exploratory efforts in the area. Such efforts were discontinued, because they were deemed economically infeasible, but the community saw the potential for Au extraction, so they began engaging in illicit mining. Illegal miners came from all over Indonesia until 2007, including Bengkulu, Ambon, Tasikmalaya, and the surrounding Bunikasih regions. From 2007 to 2019, only illegal miners from Bunikasih Village and the nearby Pangalengan Region were in charge of the mining operations in the research area. In 2019, approximately 20 groups worked on the Cibaliung River. Six to ten people formed one group, and teenage boys who could not continue their schooling participated in the mining process.

Tea farms generate 30,000 IDR (2 USD) per day per individual. More than one member of each household may work on tea plantations. A family’s monthly income ranges from 1,500,000 IDR (98 USD) to 3,000,000 IDR (197 USD). Typically, this salary is insufficient, particularly for education and healthcare. While working as miners, one family may earn between 5,000,000 IDR (328 USD) and 7,000,000 IDR (460 USD) per month per a single miner. The mining procedure does not need large amounts of money. Hg may be reused and shared with other groups. According to statistics from the village head and conversations with certain individuals, most of the residents finished elementary school, followed by those who reached junior high school and those with no formal education. This is related to the residents’ jobs; most of them are unemployed or working as household helpers and farmers.

## 2. Materials and Methods

In this study, (1) we examined the change in the residents’ values regarding the effect of ASGM on the environment and their health after conducting the medical check on the miners for ASGM-impacted health disorders, a group discussion, and a seminar using a questionnaire survey and interviews. (2) We initiated the transdisciplinary community of practice (TDCoP) activity in collaboration with the locals in order to implement alternative livelihoods as an approach to solve the ASGM problem. (3) We performed an environmental impact assessment study using tea leaves to investigate the area’s extent of exposure to toxic elements (Hg, Pb, and As) before and after the government’s efforts to stop the ASGM activities in the study area. We disclose that the shutdown of ASGM activities in this study area is not related to our study but is conducted by the government according to their plan.

The overview of the present study and events is mentioned in the following flow chart (Figure 1).

As described in Figure 1, following the preliminary study, we conducted following activities during 2019 and 2021: (1) a medical checkup of illegal miners with health concerns by medical professionals to evaluate the occupational hazards of ASGM- and Hg-related health disorders, (2) a movie viewing session and group discussion on an educational movie about Minamata Disease that occurred in the City of Minamata, Kumamoto Prefecture, Japan, in 1956, which provided information on the environmental and health impacts of Hg, (3) a seminar on the environmental and health impacts of ASGM, (4) questionnaires and deep interviews to detect changes in the views of the residents, (5) environmental impact studies by analyzing tea leaves collected at different distances from the source of ASGM conducted before and after the closure of the mines, and (6) co-designing the TDCoP of coffee plantation and beekeeping, which were chosen as alternative livelihoods by ASGM miners during discussion.

The questionnaires and deep interviews were conducted before and after transformative learning, such as watching an educational movie and seminar, in Study 1 and Study 2. The written and spoken explanation of the study to the respondents was conducted, and informed written consent was obtained prior to the study.

The medical checkups by medical doctors included the history and physical examinations of six miners who presented their present and past health concerns, present and past medical history, and signs and symptoms of the occupational hazards of ASGM and acute and chronic Hg poisoning. We requested hair samples of the participants who showed abnormal findings during their medical checkups and analyzed these samples for contents of Hg and other toxic elements.

### 2.1. Questionnaire Survey and Deep Interview

#### 2.1.1. Seminar on the Environmental and Health Effects of ASGM

A video about Minamata Disease and the environmental and health effects of ASGM was shown to the respondents. The video had a duration of 10 min and had audio and subtitles in Bahasa. The seminar was conducted in the local government office of Bunikasih Village, which is approximately 16 m^2^.

#### 2.1.2. Small Group Discussion

A discussion was conducted with the village head, public figures related to the current state of ASGM in Bunikasih, and other stakeholders (a natural resources conservation agency (BKSDA), police, forestry police, and military) regarding the cessation of ASGM activities in the study area.

#### 2.1.3. Deep Interview

Deep interviews and discussions were held with the illegal miners to (1) inform them of the results of their medical checkups, (2) investigate the residents’ changes in values and motivation to stop their ASGM activities, and (3) codesign and coproduce TDCoPs, especially to introduce alternative livelihoods to replace ASGM in the area.

#### 2.1.4. Questionnaire Survey

The questionnaire was approved by the ethical committee of the Research Institute for Humanity and Nature (RIHN) and divided into three parts: Q1 (basic information: family members, education, and type of job); Q2 (change in values before and after the seminar); and Q3 (change in values and motivation to stop ASGM activities). A paired *t*-test was applied to see the significant difference of the data for Q2 and Q3, and *p* < 0.005 was defined as significant.

### 2.2. Environmental Impact Study

#### 2.2.1. Location

The village of Bunikasih is approximately 50 km south of Bandung City, included in the Bandung District, Pangalengan Subdistrict (Figure 2a). The noted ASGM activities were on the Cibaliung River, which is behind the village of Bunikasih (Figure 2b). Sampling was conducted during two periods: Period 1 on 25 March 2019, when the ASGM activities were still occurring, and Period 2 on 9 January 2021, after the ASGM activities were almost stopped.

#### 2.2.2. Sampling Methods

Old tea leaves were obtained as samples at the same five locations between the two periods, with tea leaf heights of 40–60 cm from the ground. Samples were then acquired farther away from the location of ASGM activities and closer to the residents’ housing. In addition, the sampling positions had different surface heights, because they were on a hillside. The distance and elevation of the sampling points were obtained from Google Earth Pro data, whereas the Hg data in Period 1 were from [18]. Samples were cleaned and dried before being transported to the laboratory in RIHN.

#### 2.2.3. Analysis

The analysis of the hair and tea leaf samples was performed according to the manual of samples developed by the Japanese Ministry of the Environment [34]. Each crushed and powdered tea leaf sample (100 mg) was precisely weighed and placed in a Teflon container. Then, 1 mL of HNO_3_ (ultrapure analytical reagent, Tama Chemical Co., Ltd., Kanagawa, Japan) was poured into the container, and the mixture was heated on a hot plate at 200 °C for three to five days. Afterward, it was allowed to cool, distilled water was added to fix the volume, and the resulting solution was utilized as the sample test solution. Another 1 mL of HNO_3_ was added into a Teflon container (blank) to obtain a blank test solution. The subsequent steps followed those in the above preparation procedure (for the sample test solution). The total Hg in the sample was analyzed using a reducing-vaporization Hg analyzer (RA-43000, Nippon Instruments Co., Ltd., Osaka, Japan) according to a Hg analysis manual (Japanese Ministry of the Environment, Tokyo, Japan). The concentrations of other heavy metal elements were analyzed using inductively coupled plasma–mass spectrometry (ICP–MS; 7500cx, Agilent Technologies, Inc., Wilmington, DE, USA).

## 3. Results

### 3.1. Medical Checkup

The results of the medical checkup of six miners conducted on March 2019 revealed that they had tuberculosis (TB). TB is a common disease among miners, and its effects have been reported in previous studies [35,36]. Among the six miners, three agreed to the analysis of Hg and other toxic elements, and their total Hg contents were 1.56, 3.12, and 3.73 ppm, which are well above the safe level of Hg (1 ppm) according to the German Human Biomonitoring (HBM) Commission standard [37]; other toxic elements were within their safety ranges.

Before the researchers returned to the group discussions on July 2019, the six ASGM miners with TB had passed away.

### 3.2. Questionnaire Survey and Deep Interview

Table 1 summarizes the results of the basic information survey. Q1 was conducted for both studies; the first was conducted in 2019 before ASGM was stopped and the second in 2021 after ASGM was stopped.

A total of 46 and 51 respondents participated in Study 1 and Study 2, respectively. The former comprised 37 males and 9 females aged 37.6 ± 14.4 years and 41.7 ± 11.94 years, respectively, and the latter consisted of 19 males and 32 females. The majority of the respondents’ educational background was elementary school level, and most of the respondents lived with their families. Regarding their status of occupation, in Study 1, the majority of the respondents worked as illegal ASGM workers, which represented 30% of the total respondents, 21.7% worked as daily workers, 15.2% as full-time workers, 15% in agriculture, 6.5% in fisheries, 8.7% were housewives, 4.3% in other jobs, and 4.3% had no occupational information. In Study 2, the majority of the respondents worked as daily workers or 20% of the total number, followed by agriculture, full-time workers, and housewives. In both studies, some respondents had more than one job (i.e., some worked in ASGM and agriculture). The participants of Study 1 and Study 2 were different individuals, except for one miner.

#### 3.2.1. Change in Respondents’ Values

The changes in their values caused by the results of the medical checkups, video seminars for Minamata Disease, environmental and health effects of ASGM, and discussion on alternative livelihoods were captured using two main questions before and after conducting the seminar and deep interview: (Q2) Do you know the relation between Minamata Disease and Hg? (Q3) Do you want to join us to stop ASGM activities? The results are described in Table 2.

The responses to Q2 show that the seminar educated the respondents about the environmental and health effects of ASGM. Before the seminar, 89.1% of the respondents of Study 1 and 84.3% from Study 2 did not know about Minamata Disease and the other dangers of Hg. However, after the video seminar, 21.7% and 13.7% of the respondents of Study 1 and 2, respectively, knew them well, and 10.9% and 58.8% of the respondents of Study 1 and 2, respectively, knew enough about Minamata Disease and the other harmful effects of Hg (Table 2). The results were statistically significant in the paired *t*-test.

As for Q3, the respondents expressed having a change in values or good motivation to stop ASGM activities. In Study 2, before the study, 41.2% of the respondents had doubts about stopping ASGM. However, after the study, 88.3% of the respondents agreed and 7.8% strongly agreed to stop ASGM activities.

#### 3.2.2. Closure of Illegal ASGM

The Indonesian government has made many attempts to stop ASGM activities in Bunikasih. In August 2019, several government agencies confiscated and destroyed ASGM facilities.

After counseling the illegal miners and residents of Bunikasih regarding Hg and its hazards, along with the medical checkup findings, members of the community became willing to cease their mining activities and start other careers or continue to work as plantation laborers. Furthermore, transdisciplinary studies were conducted on the Bunikasih community to stop its ASGM activities.

#### 3.2.3. Process of Transdisciplinary Community of Practice (TDCoP) to Form Alternative Livelihoods

Through the deep interviews with illegal miners and residents in 2021, a group of TDCoPs was formed with 34 participants who were willing to stop working in ASGM. According to the survey, the most common reasons for their motivation were their concern for their health if they continue working in ASGM, little results from mining, and concern for their future. Consequently, the participants started dialogs for implementing alternative livelihoods, such as tea and coffee planting, vegetable farming, goat farming, and beekeeping. Among all of these, coffee planting and production and beekeeping were finally chosen by the participants as alternative livelihoods for TDCoPs.

### 3.3. Environmental Impact Study

The Hg, Pb, and As contents of the tea leaf samples from the two sampling periods are in Table 3.

The Hg content of the tea leaf samples in Period 1 was 0.42–0.72 mg kg^−1^, whereas that in Period 2 was 0.04–0.15 mg kg^−1^. The Pb concentrations were 0.58–0.79 and 0.21–0.56 mg kg^−1^ in Periods 1 and 2, respectively. The As content was 0.21–0.42 mg kg^−1^ in Period 1 and ND–0.14 mg kg^−1^ in Period 2. Overall, the Hg, Pb, and As contents decreased from Period 1 to Period 2.

## 4. Discussion

### 4.1. Changes in Values of Illegal Miners and the Formation of Transdisciplinary Community of Practice

This study was conducted as part of the Sustainable Regional Innovation for Reducing Risk of High-impact Environmental Pollution (SRIREP) initiative of the RIHN in supporting TDCoPs [18] to address the issue of ASGM, which is rooted in poverty. The SRIREP Project has formed several TDCoPs with locals in Gorontalo Province, Indonesia, to solve different problems [38]. TDCoPs are groups of individuals who collaborate to develop and practice problem-solving. These organizations include local citizens, administrators, merchants, and other nonacademic individuals. The effectiveness of their endeavors is contingent on their capacity to engage those who are uninterested in the problems to be handled. According to Wenger et al., TDCoPs aim to gather individuals who are interested in cross-border problems and help people gain new knowledge. This holistic community development operates via three realms: action/application, dialog/process, and self-development/reflection [39]. Participation has a multi-membership nature, with participants belonging to official organizations and TDCoPs and collaborating to produce knowledge rooted in the field through a dialog between the two parties. This knowledge is then reflected in the activities of the official organizations, resulting in social implementation.

According to Wenger et al., TDCoPs progress through five stages: potential, coalescence, maturity, activity, and transformation [39]. In this study, after the initial dialog with the residents and illegal miners on the environmental and health impacts of ASGM, especially on the multiple TB cases in miners, seminars, and alternative livelihoods, the illegal miners experienced a change in their values and decided to stop illegal ASGM activities. In this case, the educational video is referred to as transformative learning, and the health impacts can be defined as a transformative boundary object, which is a concept that promotes collaboration and knowledge creation in TDCoPs. Based on this knowledge, TDCoPs were formed for exploring sustainable alternative livelihoods, such as coffee production and beekeeping (Figure 3).

TDCoP employs an unranked structure in which members can freely state their opinion regardless of positions, in contrast to the traditional Asian hierarchy involving the top-down approach. However, one of its disadvantages is that it is an informal organization. Currently, the TDCoP of coffee production has shown positive results, although the income is still insufficient for fulfilling the needs of participants, while beekeeping TDCoP has also encountered some challenges. In the future, these TDCoPs will be evaluated for productivity and sustainability. Additionally, because ASGM has been practiced in the study area, the analysis for heavy metal contamination in coffee beans should be considered in the future to ensure food safety.

There is a lack of awareness among ASGM miners regarding the health and environmental risks of ASGM [25], and poor awareness has also been reported among miners and nonminers regarding the health and environmental impacts of Hg [26,27,40]. This calls for policy interventions regarding awareness programs on Hg and ASGM. Although educational and technical interventions have resulted in improved awareness on Hg reduction among ASGM miners, these attempts are not expected to exist for the long term or permanently because of the high operation costs and miners’ indifferent interests. As a result, miners may go back to the Hg amalgamation method [19]. However, herein, we employed an approach different from those of previous educational studies. We employed TDCoP, which permits a two-way dialog with the respondents, closer interactions with them, and longer intervention periods.

A knowledge–action gap to reduce Hg in ASGM has been reported [19,28,29,30]. This implies that miners and community members often report an interest in Hg reduction due to the reflection of social desirability bias, but they do not take action in reality. The current studies also face this challenge. In addition, in the Bunikasih case, there is the risk of a lack of other options besides seeking alternative livelihoods due to the closure of ASGM in villages even when they want to continue to engage in ASGM. This questions our interpretation of their changes in values concerning Hg and ASGM. However, according to our observations, the miners in Study 1 did not seek out ASGM jobs in areas other than Bunikasih, even though ASGM miners may go back to mining, as mentioned in previous studies. This confirms our interpretation that there is a change in their perception towards ASGM.

An awareness of the dangers of Hg and changes in values concerning Hg and ASGM in response to transformative learning were gained by illegal miners and residents; however, according to the previous reports on the failure of various attempts, the results in our study might also be doubted as only a short-term effect of educational interventions. However, in our study area, the chances of the communities falling back into Hg usage and ASGM practices are low for the following reasons. First, the fatal risk of ASGM from TB has been observed by illegal miners and residents. Second, the government (police and forest rangers) strictly enforces the complete cessation of ASGM in villages. Third, the ASGM results of the study area are not promising. Lastly, there is potential autonomous management of the livelihoods of the villagers through TDCoP. Nevertheless, there is still a need for further in-depth study on the people’s behavior. In addition, this approach can be generalized by collecting case reports, but it has limitations. For example, it can only be applied to an area that has a situation similar to what exists in the present study area, where nonmigrating ASGM is practiced. Moreover, there is a need for government control and provision for alternative sources of livelihood, such as plantations.

### 4.2. Environmental Impact Assessment: Correlation to Distance, Elevation, and Sources of Pollution

Hg dispersal can occur up to 10 km [19] and at different elevations [4]. This was also shown in Bunikasih; the Hg contents in Periods 1 and 2 decreased with an increase in the distance and elevation differences from the ASGM activity location. The impact of the Hg distribution decreased at a 77–141 m distance and an elevation difference of approximately 20 from the ASGM activity location [18] (Figure 4a,b).

The distributions of Pb and As provide additional information. The Pb concentration was higher at the ASGM and residential areas. This leads us to conclude that the source of Pb pollution is anthropogenic [3]. In contrast, the As concentration seemed random, showing that the source of As pollution could be natural and/or anthropogenic. Hg, Pb, and As are heavy metals with a high affinity for dissolved hydrogen sulfides and chloride in hydrothermal ore solutions and are commonly associated with gold ores. Therefore, they can be released into the environment from the ore during mining processes. Additionally, the diffusion of heavy metals through the air is influenced by the wind direction, uptake ability of plants, and plant parts [41].

### 4.3. Contamination Level and Environmental Assessment of ASGM Activity Closure

The data distribution in Table 4 shows that the Hg contamination level in the tea leaves decreased significantly; the mean value of the Hg data decreased from 0.54 ± 0.14 to 0.08 ± 0.04 mg kg^−1^ from Period 1 to Period 2. The Pb and As levels also decreased. The mean Pb value changed from 0.66 ± 0.09 to 0.34 ± 0.14 mg kg^−1^, and the mean As value changed from 0.34 ± 0.12 to 0.07 ± 0.06 mg kg^−1^.

In general, the Hg concentration in plant leaves is below 0.1 mg kg^−1^ [42]. Hg contents exceeding 1 mg kg^−1^ can be dangerous [43]. As for As and Pb, the normal limits in plants are 0.02–7 and 1–13 mg kg^−1^, respectively [44,45]. Therefore, the environment in Bunikasih is polluted with Hg, while the Pb and As levels are normal (Table 4). Nonetheless, the Hg, As, and Pb contents in Bunikasih are smaller than those in the *Pteris vittata* plants at the ASGM location in Gorontalo, Indonesia, which are 5.2, 17,700, and 39 mg kg^−1^ [4], respectively. The Hg content in Bunikasih is also lower than that in forage plants at the ASGM location in Bombana, Indonesia, which is 9.90 mg kg^−1^ [14].

The environmental assessment conducted after the ASGM closure efforts in Bunikasih showed significant decreases in Hg, Pb, and As contamination (Table 4). Thus, the closure of ASGM activities is going well, and illegal miners agree with these cessation efforts.

### 4.4. Limitation

This study was limited to the health assessments of villages. Hg- and ASGM-related health problems among several residents, which could have shown whether there is health improvement after banning the ASGM, were not considered.

## 5. Conclusions

In this study, the illegal miners and residents of Bunikasih, Indonesia, showed changes in their values regarding the environment and their health and expressed a willingness to stop their illegal ASGM activities, which is a fundamental element needed to form TDCoPs in the study area to facilitate regional innovations by working with residents and other stakeholders. This transformative change in their values is attributed to the awareness gained regarding their health concerns and the availability of alternative sustainable sources of livelihood. However, the approach employed herein can only be applied in an area that has a situation similar to that of the present study area, where nonmigrating ASGM is practiced. In addition, there is a need for government control and alternative sources of livelihood, such as plantations.

We also examined the environmental pollution caused by Hg, Pb, and As in the study area during the closure of ASGM activities using tea leaf samples obtained between the ASGM location and village housing during two time periods. The mean concentrations of all the elements decreased between the two sampling periods; the concentration of Hg decreased from 0.54 ± 0.14 to 0.08 ± 0.04 mg kg^−1^, that of Pb changed from 0.66 ± 0.09 to 0.34 ± 0.14 mg kg^−1^, and the As content was reduced from 0.34 ± 0.12 to 0.07 ± 0.06 mg kg^−1^. These decreases indicate that stopping the ASGM activities resulted in the rapid reduction of the environmental Hg levels in the village. The shift from ASGM was driven by government and law enforcement interventions. The TDCoP approach has sought to establish alternative livelihoods to make these changes permanent, although more time is needed to determine the long-term success of the reduction in ASGM and Hg usage.

## Figures and Tables

**Figure 1 ijerph-20-06663-f001:**
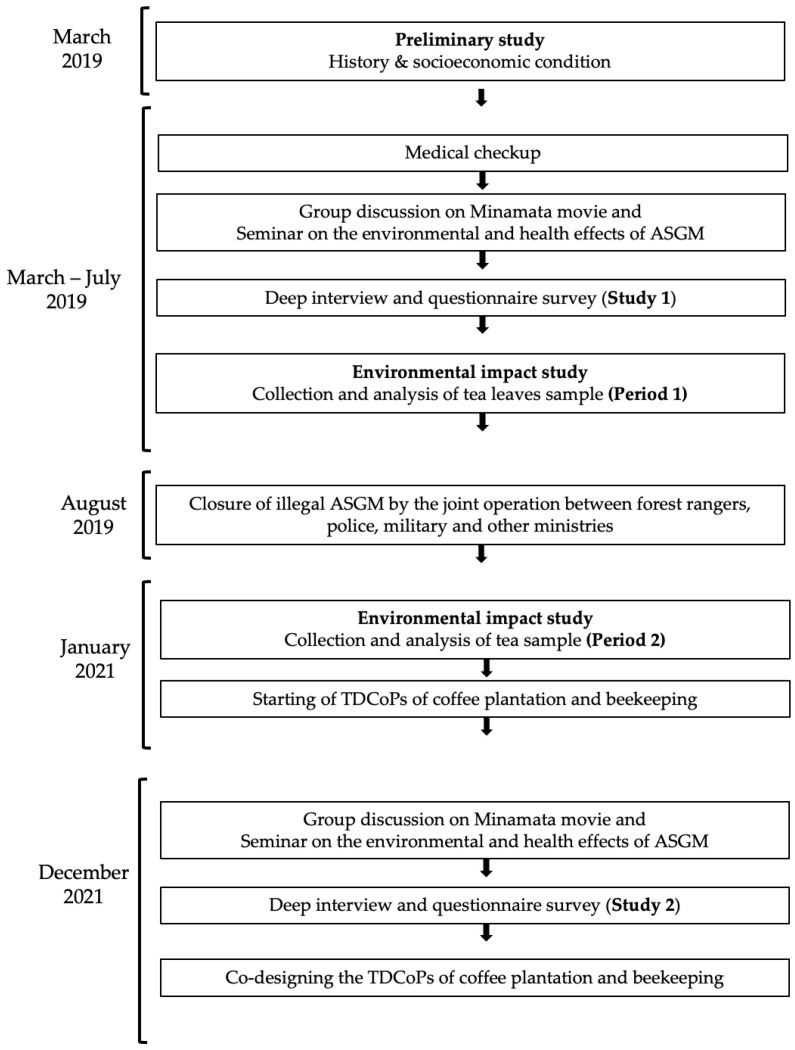
Flow chart of the study. Abbreviations: ASGM, artisanal and small-scale gold mining; TDCoP, transdisciplinary community of practice.

**Figure 2 ijerph-20-06663-f002:**
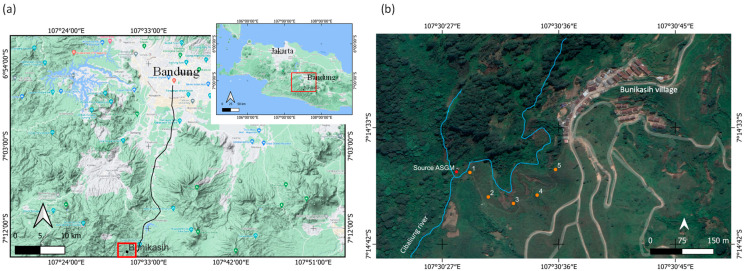
(**a**) Location map of Bunikasih, and (**b**) a sampling map of tea leaves.

**Figure 3 ijerph-20-06663-f003:**
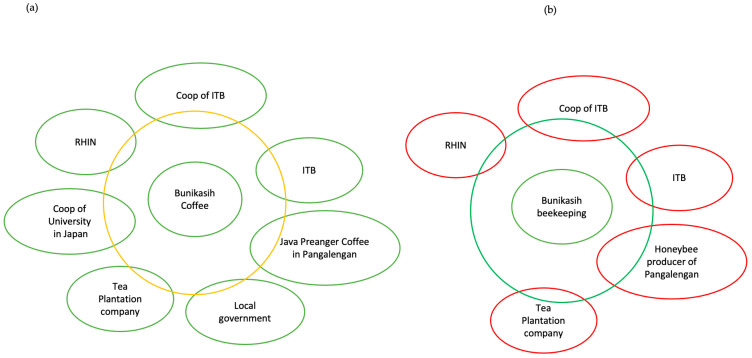
(**a**) Transdisciplinary Community of Practice (TDCoP) of coffee production and (**b**) TDCoP of beekeeping. Abbreviations: RIHN, Research Institute for Humanity and Nature; ITB, Bandung Institute of Technology.

**Figure 4 ijerph-20-06663-f004:**
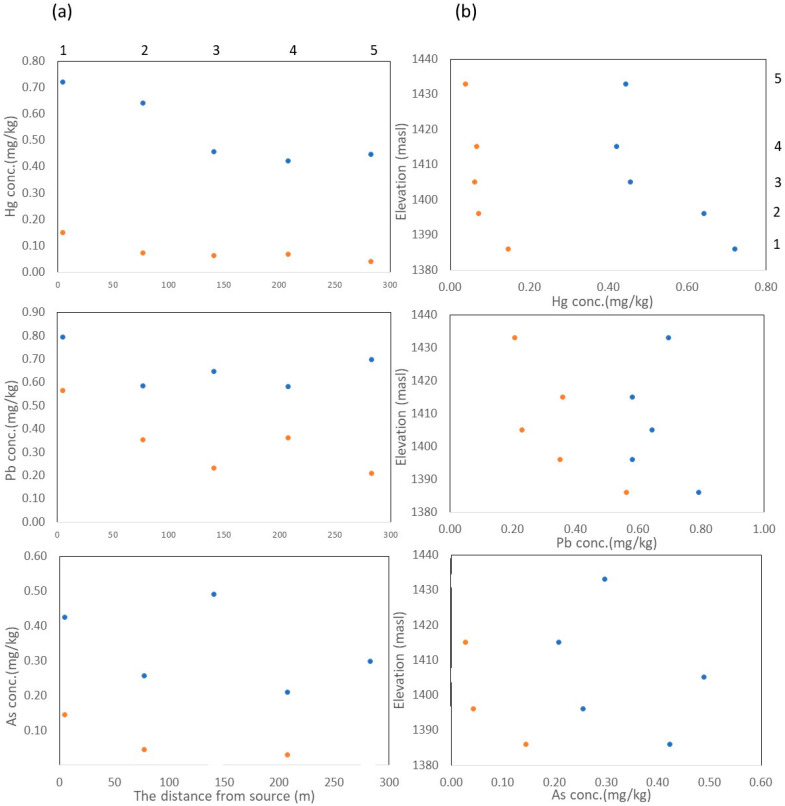
Concentrations of Hg, lead (Pb), and arsenic (As) versus (**a**) the distance from ASGM activity location (m) and (**b**) altitude (m.a.s.l.) (blue circle = Period 1; orange circle = Period 2).

**Table 1 ijerph-20-06663-t001:** Result of the questionnaire survey: Q1: basic information of the respondents of Study 1 conducted in 2019, and Study 2 was conducted in 2021.

Variable	*n*; Study 1 in 2019	*n*; Study 2 in 2021
Sex		
Male	37	19
Female	9	32
Age (years)		
Male	37.6 ± 14.4	34.1 ± 12.4
Female	41.7 ± 11.94	36.2 ± 13.5
Level of Education		
High school	3	-
Junior high school	8	15
Elementary school	28	33
Other form of education	1	-
No formal education	3	3
No information	3	-
Marital Status		
Married	36	43
Unmarried	6	7
No information	4	1
Occupation Status		
ASGM	14	1
Daily workers	10	20
Full-time workers	7	7
Agriculture	7	9
Fishery	3	-
Housewife	4	9
Others	2	2
No information	2	1

**Table 2 ijerph-20-06663-t002:** Result of the questionnaire survey: Q2 and Q3: Changes in values of the respondents of Study 1 conducted in 2019 and Study 2 conducted in 2021.

Question	Response	Study 1		Study 2	
		Before (*n*, %)	After (*n*, %)	Before (*n*, %)	After (*n*, %)
Q2: Do you know the relation between Minamata Disease and Hg? *	Not at all	44, 89.1%	12, 26.1%	43, 84.3%	1, 2.0%
	A little	1, 2.2%	13, 28.3%	6, 11.8%	13, 25.5%
	More or less	2, 4.3%	3, 6.5%	0	0
	Know well	1, 2.2%	10, 21.7%	2, 3.9%	30, 58.8%
	Know enough	0	5, 10.9%	0	7, 13.7%
	No answer	1, 2.2%	3, 6.5%	0	0
Q3: Do you want to join us to stop ASGM activities? *	Strongly disagree	0	1, 2.2%	0	0
	Disagree	1, 2.2%	0	1, 2.0%	0
	Doubtful	0	0	21, 41.2%	2, 3.9%
	Agree	25, 54.3%	10, 21.7%	24, 47.0%	45, 88.3%
	Strongly agree	18, 39.2%	32, 69.6%	5, 9.8%	4, 7.8%
	No answer	2, 4.3%	3, 6.5%	0	0

* Indicates *p* < 0.005 in the paired *t*-test.

**Table 3 ijerph-20-06663-t003:** Mercury (Hg), lead (Pb), and arsenic (As) concentrations in the tea leaf samples in Period 1 (25 March 2019) and Period 2 (9 January 2021).

Sampling Point	Hg (mg kg^−1^)		Pb (mg kg^−1^)		As (mg kg^−1^)		Distance from Source (m)	Elevation (m.a.s.l.)
	1	2	1	2	1	2		
1	0.72	0.15	0.79	0.56	0.42	0.14	5	1386
2	0.64	0.07	0.58	0.35	0.26	0.04	77	1396
3	0.46	0.06	0.64	0.23	0.49	ND	141	1405
4	0.42	0.07	0.58	0.36	0.21	0.03	208	1415
5	0.45	0.04	0.70	0.21	0.30	ND	283	1433

**Table 4 ijerph-20-06663-t004:** Distribution of tea leaf samples based on the minimum; median; mean and standard deviation; and maximum Hg, Pb, and As values.

Elements	Hg (mg kg^−1^)		Pb (mg kg^−1^)		As (mg kg^−1^)	
Period	1 (*n* = 5)	2 (*n* = 5)	1 (*n* = 5)	2 (*n* = 5)	1 (*n* = 5)	2 (*n* = 5)
Minimum	0.42	0.04	0.58	0.21	0.21	0.03
Median	0.46	0.07	0.64	0.35	0.30	0.04
Mean ± SD	0.54 ± 0.14	0.08 ± 0.04	0.66 ± 0.09	0.34 ± 0.14	0.34 ± 0.12	0.07 ± 0.06
Maximum	0.72	0.15	0.79	0.56	0.49	0.14

## Data Availability

Not applicable.

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
