# Peer review of "Change in Values of Illegal Miners and Inhabitants and Reduction in Environmental Pollution after the Cessation of Artisanal and Small-Scale Gold Mining: A Case of Bunikasih, Indonesia"

_ijerph, 2023, doi:10.3390/ijerph20176663_

Round 1
Reviewer 1 Report
Summary:
Artisanal and small-scale gold mining (ASGM) is an environmental hazard affecting the local economies and living conditions in developing countries in the world, in large part due to limited employment opportunities outside of this industry in areas where this mining is conducted. This study aims to understand the social factors affecting individuals’ decisions to pursue ASGM, provide education regarding health and environmental hazards of ASGM, and observe how community perceptions of ASGM and the environmental impacts changed following discontinuation of ASGM over a two-year period. The results demonstrate increased community knowledge of ASGM environmental and human health impacts concurrent with discontinuation of ASGM practices, and these events parallel decreases in environmental pollution of heavy metals during the same period.
Overall Comments:
The authors are commended for this extremely interesting manuscript with a variety of strengths, including engaging the community as stakeholders in research (community-based participatory research), the acknowledgement and discussion of environmental justice as impacted by socioeconomic issues, and the pre-/post- evaluation of environmental pollution following a notable change in human impact on the environment (i.e., the discontinuation of ASGM). The use of transdisciplinary communities of practice (TDCoPs) is a particularly interesting implementation in this study to understand and address the problems of ASGM within the larger framework of social determinants of health. The manuscript overall is extremely well-written, and it is straightforward to follow the progression from their “preliminary study” (though further context on that preliminary study would be appreciated) to their current study. Presumably, this work has the potential to be reproduced and impact other areas in which ASGM is still prevalent due to similar social factors. The references appear to be appropriate to the topic with regard to TDCoPs, and ASGM-related environmental hazards.
In particular, the manuscript is outstanding in terms of its analysis of Hg, Pb, and As in tea leaves as representative of environmental contamination, as well as the thorough discussion of the results of these findings in terms of understanding the sources of Hg, Pb, and As, which were all different environmental sources, as well as how the observed values are related to the reference ranges of expected values in environmental sources like plants.
Although this is a very strong manuscript and would be a very good contribution to the literature, there are some ways in which improvements could be made to make this manuscript even stronger, and some ways that should certainly be addressed to ensure clear interpretation of the data within the context of this study.
The broad specific comments below address generally a few points that should be evaluated by the authors even beyond the individual specific comments listed thereafter.
Broad specific comments:
1) The “medical checkup” in 2019, its intent, and its results not well-discussed, though it is stated and implied multiple times in the manuscript that the medical checkup had some impact on the results of the survey. These points are addressed in the individual specific comments below.
2) The “preliminary study” before the current one is mentioned a few times, but with little context; further context would be very helpful to understand whether the preliminary study may have affected the current study’s results. Further on this point is addressed in the individual specific comments below.
3) The existing Discussion is good, but could benefit from some additional points being addressed:
a) Few if any limitations to this study are discussed. Is this study generalizable; why or why not?
b) Potential areas of bias are not discussed well. How did concomitant environmental health/pollution work (as implied by parallel decreases in Pb and As with what is observed with Hg) affect these results?
4) There are some minor capitalization issues, such as reference to “Study 2” in places (e.g., p.6) versus “study 2” in other places (e.g., p.7), or “Period 1” versus “period 1”.
Individual Specific Comments:
p.2:
“Various samples, such as air, water, sediment, and soil, have been used to measure the heavy metal pollution caused by ASGM activities [3][4][6][7][13][14]. Many factors influence the distribution of pollution, including its natural origin (e.g., rainfall and wind) and human activities (e.g., duration of refining and proper/improper disposal of Hg waste) [3,6]. Hence, pollution spread, level, and behavior in areas of ASGM activity should be determined.” Recommend further discussion of environmental measurements of these samples in Indonesia and/or in the local area of this study; if such sampling has not been conducted previously, then that should be stated.
p.2-3:
Section 1.2 “Transdisciplinary Research Practice to Tackle ASGM Issues”. There is no problem with this section, but wanted to point it out as a strong section in which commentary was presented on environmental justice in terms of the intertwined issues with poverty.
p.3:
Section 1.3 “Preliminary Transdisciplinary Study at Study Site”. Appreciate the discussion of economic motivations of pursuing ASGM in context of local social factors. Although the authors do a very good job of providing the context for the stated financial numbers, such as ranges of family income in Indonesian Rupiah (Rp), would recommend putting a different currency as context (e.g., USD or EUR) in addition to Rp for international readers (such as myself).
p.3-4:
Regarding the “preliminary study” discussed, were these results published? There are not citations within this section of the manuscript that suggest the results were published, so perhaps they were not. It may be worth mentioning some additional context of the preliminary study. It also seems possible that there may be some effect of the preliminary study work on the results of the present study, with respect to the discussed counseling of “the residents, including the miners, regarding Hg and its hazards”. Additionally, given that this topic is of interest and/or concern to the researchers, it may be possible that stakeholders and participants were influenced by the preliminary study when the work proceeded to the current study. These points may merit comment in the Introduction or Methods with respect to the purpose/context of the preliminary study, which the points in Section 1.3 seem to imply the purpose was to understand the history and context of ASGM in the local area, as well as to provide some education. These points could then also be mentioned in the Limitations in terms of the potential for bias with respect to the work done in the preliminary study. If it is known, it would be worthwhile to comment on the number of individuals that had the education from the preliminary study and were also part of the current study, again, in case this may imply some bias (even if small).
p.4:
With regards to the “medical checkup” described as part of this study, what was the purpose of this medical evaluation? To determine existing health conditions? To do biomonitoring of mercury (or other metals)? To gain trust of the participants? It would be worth commenting on both why these evaluations were performed as well as (even briefly) what information was ascertained. This point appears more relevant later in the manuscript (p.7), where the following text reads: “The change in values caused by the result of medical checkup, video seminars for Minamata Disease, environmental and health effects of ASGM and discussion on alternative livelihoods was captured using two main questions before and after conducting the seminar and deep interview”; again, from this text, there are implications about the effect and impact of the “medical checkup”, but there is not enough information provided to the reader regarding what the medical evaluation entailed, nor does the p.7 text from above suggest that the medical evaluation was related to Q2 or Q3. Specifically, the medical evaluation was not mentioned in either question, although it is implied that there was some education or discussion that must have occurred as part of that evaluation, but whatever action occurred as part of that evaluation is not discussed in this manuscript. The impact of the medical checkup was again referenced on p.8, “After counseling the illegal miners and residents of Bunikasih regarding Hg and its hazards along with the medical checkup findings, members of the community became willing to cease their mining activities and start on other careers or continue to work as plantation laborers”; in this instance the “findings” of the medical evaluation was referenced vaguely, but such findings (even in terms of what was tested, discussed, or evaluated) are not mentioned in this manuscript. Section 3.1.3, in which a motivation for discontinuing ASGM was noted regarding the miners’ “concern for their health if they continue working in ASGM”, also implies that some information was given as part of the medical evaluation that was relevant to the participants’ decision-making.
p.4, Figure 1: The second box referencing “Deep interview and questionnaire study” (2021) is labeled as “Study 1”, though likely should be labeled as “Study 2” (as it is referenced later in the manuscript).
p.6:
Results, Section 3.1:
p.6-7, Table 1:
a) Typo in caption of Table 1: “conduced”.
b) Recommendation: rather than repeating the subcategories of each data element, e.g., under “Sex” listing “Male” and “Female” twice, the table could be laid out with the higher-level category (e.g., “Sex”) followed by each subcategory slightly indented, where then the numbers within the table can be displayed on their own for more easy readability (and less text overall).
p.11, Table 4:
Table 4 demonstrates significant changes in Hg, Pb, and As from 2019 (ASGM ongoing) to 2021 (ASGM discontinued). The text on p.11 (above the table) suggests that the 2019 Hg results were abnormal, while the 2021 Hg results, and all of the Pb and As results were normal, and a reasonable explanation is provided; i.e., Hg has ASGM as its source, Pb is related to human activity, and As is related to likely multiple sources. However, there still appears to be some notable decreases in Pb and As comparing 2019 to 2021; since the sources of these are not presumed to be related to ASGM, can the authors comment on why those decreases have occurred? Are there other environmental advocacy, education, or legislative efforts occurring in the same local area during the same time period? Whatever the reason, if it can be identified (or at least supposed), would this effect be expected to be contributing to the decrease in Hg from 2019 to 2021, even presuming that some of that effect is related to the discontinuation of ASGM?
The quality of the English language in this manuscript appears to be at or very close to the level of a fluent English speaker/writer.
Reviewer 2 Report
Overall comments:
This is a strong study that can be publishable following revisions. The before-and-after results of Hg levels as measured in/on tea leaves is particularly noteworthy, as few longitudinal studies exist of mercury levels before and after ASGM reduction in a single site. Indeed, I would encourage the authors to consider publishing those results as a stand-alone paper, splitting them out from the educational / TDCoP element of this paper. Most of the material relating to the leaf sampling is already strong, though see some specific comments below. The most important improvement for this part would be to make clear whether the analysis is measuring mercury deposited on the surface of the leaves, incorporated into the plant matter, or both, as this significantly impacts the interpretation of results.
The second component (educational / TDCoP element) is also interesting, though less novel than the environmental sampling result, given the large literature on mercury reduction interventions around ASGM, including educational efforts, studies of miners’ perceptions, and alternative livelihood efforts. The authors need to engage with that literature much more extensively, because it calls into question the optimistic interpretation the authors give to their results. Notably, in lines 316-319 they admit that the alternative livelihoods intended to replace ASGM have not produced the desired results – relating this finding to the literature, it is highly likely that people will revert to ASGM and mercury use once the crackdown ends, the intervention ends, or savings and patience run out. This component of the paper is still publishable and interesting, but needs a much stronger integration with the literature, as outlined in the following paragraph.
The current literature review (in the introduction) focuses on environmental and health impacts of mercury. It is indeed important to note these facts, but in the context of the current study, I find this to be the wrong literature review. This material can be summarized succinctly in a single paragraph. In turn, much more needs to be said about the state of the field of interventions to reduce or eliminate mercury in ASGM. There is an extensive literature on this topic that the paper does not engage with seemingly at all. Importantly for this study, there is strong evidence that interventions of multiple styles usually fail and/or produce fleeting results (see Veiga and Fadina 2020; Zolnikof and Ortiz 2018). This literature is highly relevant for the current study, which reports short term results of an intervention – i.e., the literature calls into doubt whether the current results should be interpreted optimistically, in light of the well documented failures of similar interventions even when immediate results seem promising.
Furthermore, there are numerous studies specifically about education and awareness raising about mercury in ASGM settings, which also are not engaged by the paper. See the list below of sources to consult – and note, this list is far from exhaustive. For me, it is impossible to accept a paper that so clearly fits within this literature, but does not acknowledge or engage with the literature. The authors must situate their own work within the broader body of work to show what is novel, interesting, and important in their work (and to allow more realistic interpretation of the results). The TDCoP approach can be seen as a response to the frequent criticisms of top-down mercury interventions, but this line is not drawn in the paper as written.
Key reviews regarding difficulty of lasting Hg reduction or elimination in ASGM:
- Veiga and Fadina 2020 DOI 10.1016/j.exis.2020.06.023
- Zolnikov and Ortiz 2018 DOI 10.1016/j.scitotenv.2018.03.241
Prior studies engaging with education of ASGM miners and communities about Hg, and/or perceptions of mercury:
- Ottenbros et al 2019 DOI 10.1016/j.envint.2018.10.059
- Sana et al 2017 DOI 10.11604/pamj.2017.27.280.12080
- Charles et al 2013 DOI 10.1186/1471-2458-13-74
- Jonsson et al 2013 DOI 10.1016/j.resourpol.2012.09.001
- Nkuba et al 2019 DOI 10.1016/j.jclepro.2019.01.174
- Quispe Aquino et al 2022 DOI 10.1016/j.envres.2022.114092
- Wireko-Gyebi et al 2020 DOI 10.1080/10803548.2020.1795374
- Aram et al 2021 DOI 10.1016/j.resourpol.2021.102108
- Armah et al 2016 DOI 10.3389/fenvs.2016.00029
- Cuya et al 2021 DOI 10.1016/j.gecco.2021.e01816
- Velez-Torres et al 2018 DOI 10.1177/1070496518794796
-
Specific comments:
- Line 54-56: Could reference key document here regarding differences in mercury pollution from whole ore amalgamation vs. amalgamation of concentrates – Yoshimura et al 2021 DOI 10.1007/s40831-021-00394-8
- Section 1.2: The first few sentences here are superfluous, giving vague explanation of transdisciplinary research, which is already the norm in work around ASGM and need not be explained. In turn, the specific concept of TDCoP is insufficiently explained. I would recommend reversing the emphasis, reducing the general statement of the need for transdisciplinary research to a single line, and expanding the discussion of the specific TDCoP concept to explain more fully and situate the research. (*Note – The start of section 4.1 includes the type of material that I would like to see in Section 1.2)
- Line 124: Typo, should say 1,500,000 to 3,000,000 Rp (missing a 0 – currently says 3,00,000)
- Line 125-126: specify whether this amount is for a single miner, or multiple miners within the family
- Line 132-141: This paragraph should be part of Section 2 Materials and Methods, as it describes the current study (in contrast to the rest of section 1.3, which seems to describe prior research and the study site)
- Figure 1: Why are dates given for all other activities, but not the preliminary study? Add date for preliminary study
- Figure 1: Should the interview and survey box in December 2021 (bottom) say Study 2, instead of 1?
- Figure 1: I am very curious – was it just good luck that the research team did work in March to July 2019, immediately preceding the August closure of illegal ASGM, or did the team know in advance of this pending intervention and specifically target the research to allow a before and after design? If the latter, describe as such.
o Related to this – section 2.1.2 leaves me wondering, is it even possible that this research intervention actually caused the closure of illegal ASGM by informing local authorities and law enforcement about the issue? If that is the case, there would be significant ethical issues to consider.
- Line 157: beekeeping and coffee were chosen by whom to be alternative livelihoods? Chosen by miners / community, or by outsiders? Based on what criteria?
- Section 2.1.4: The survey is described as measuring change in values, but the description of methods does not make clear whether the surveys were actually collected before and after respondents watched the film / engaged in the seminar. Need to be explicit on this point.
- Section 2.2.3: No citations are provided to justify this analysis approach (especially the extraction method from the tea leaves) – is this a novel method or an established method? If established, cite. If novel, explain why this was used, and if alternative methods exist.
- Line 223-224: Why the big change in gender from study 1 to study 2? Were the same population groups targeted each time?
- Line 250-253: There is a well documented knowledge-action gap for mercury reduction in ASGM (see Aram et al., 2021; Armah et al., 2016; Cuya et al., 2021; Veiga and Fadina 2020) – meaning, miners and community members will often report an interest in mercury reduction but then not act on it. This could be a reflection of social desirability bias (telling researchers what they think the researchers want to hear) or finding alternatives inviable. It is important to acknowledge this challenge, and explain (here or elsewhere) how it impacts your interpretation of the current study.
- Section 3.1.2: As noted above, I would like to hear explicitly the relationship between this study and the government interventions. Did the researchers know in advance of the coming crackdown? Did the research itself help catalyze the crackdown on ASGM? Are there any ethical concerns, given the livelihood importance of ASGM?
- Line 276: See note from line 157 – chosen by who? And are any results available regarding the outcome of this change? That is, have community members been able to garner sufficient incomes from coffee, beekeeping, etc. that can make the shift away from ASGM sustainable in the long term? Or are many likely to return to ASGM if incomes in these alternative activities are lower? Again, this relates to the literature around Hg-reduction efforts in ASGM worldwide, which have often shown short-term results but in the long term reversion to ASGM and Hg use.
- Section 3.2 (and Section 2.2.2) – Please specify, do your methods capture Hg deposited on the surface of tea leaves, Hg incorporated into the plant material of the tea leaves, or both?
- Line 310: This statement attributes cessation of ASGM activity to miners’ “change in their values”, but elsewhere you note that the government and police shut down ASGM. Did miners really give up ASGM due to a change in values, or were they forced to give up ASGM and only after the fact attributed this to a change in values?
- Line 316-319: See my comments on Lines 157 and 276. It is curious to me that you have chosen, earlier in the paper, to frame beekeeping and coffee as viable alternative livelihoods, when here you acknowledge that both have shown limited results in practice. Knowing this outcome, I find it somewhat disingenuous to present them as solutions elsewhere in the paper. Interpreting the results in these lines with reference to the broader literature on mercury ASGM interventions, I would expect that a return to ASGM and mercury use is highly likely for these participants.
- Line 329: Where does this figure of 280-300 m come from? Looking at Table 3 and Figure 4a, it appears that there is an obvious trend with points 1 and 2 having higher mercury levels, and points 3, 4, and 5 having similar, lower mercury levels. The blue period 1 trend line drawn on the graph (4a) seems inappropriate, as there appears to be a phase shift going on – should not use a linear trend (should also consider spacing the points according to meters distance, instead of equidistant on the chart). In this light, it seems that the true fall-off in Hg contamination occurs between 77 and 141 m (between points 2 and 3).
- Section 4.3: This is a very solid and important finding, that Hg levels decreased significantly over a short time when ASGM stopped. (But also, see note re: Section 3.2 / 2.2.2 – are you measuring mercury on the leaf surface, incorporated in plant material, or both?)
- Line 361: typo, “wiliness” should be “willingness”
- Line 370-371: I take issue with this interpretation. What I would suggest: These decreases indicate that stopping ASGM activity resulted in rapid reductions in environmental mercury levels in the village. The shift away from ASGM was driven by government and law enforcement interventions; the TDCoP approach has sought to establish alternative livelihoods to make these changes permanent, though more time is needed to determine the long-term success of ASGM and mercury use reductions.
Round 2
Reviewer 2 Report
I am satisfied with the revisions and applaud the authors for engaging with the recommended literature. The revised paper is, in my view, ready for publication. There are a few small errors to resolve, such as misspelling the word "dialogue" in at least two places.